# Functional Design of Pocket Fertigation under Specific Microclimate and Irrigation Rates: A Preliminary Study

Chusnul Arif [1,*], Yusuf Wibisono [2], Bayu Dwi Apri Nugroho [3], Septian Fauzi Dwi Saputra [4], Abdul Malik [1], Budi Indra Setiawan [1], Masaru Mizoguchi [5] and Ardiansyah Ardiansyah [6]

[1] Department of Civil and Environmental Engineering, IPB University, Kampus IPB Darmaga, Bogor 16680, Indonesia; malik.abede3@gmail.com (A.M.); budindra@apps.ipb.ac.id (B.I.S.)
[2] Department of Bioprocess Engineering, Brawijaya University, Malang 65141, Indonesia; y_wibisono@ub.ac.id
[3] Department of Agricultural and Biosystem Engineering, Gadjah Mada University, Yogyakarta 55281, Indonesia; bayu.tep@ugm.ac.id
[4] Civil Engineering and Management, School of Vocational Sciences, IPB University, Bogor 16680, Indonesia; septianfauzi@apps.ipb.ac.id
[5] Graduate School of Agricultural and Life Sciences, The University of Tokyo, Tokyo 113-8657, Japan; amizo@mail.ecc.u-tokyo.ac.jp
[6] Department of Agricultural Engineering, Jenderal Soedirman University, Purwokerto 53125, Indonesia; ardi.plj@gmail.com
* Correspondence: chusnul_arif@apps.ipb.ac.id

**Abstract:** Irrigation and fertilization technologies need to be adapted to climate change and provided as effectively and efficiently as possible. The current study proposed pocket fertigation, an innovative new idea in providing irrigation water and fertilization by using a porous material in the form of a ring/disc inserted surrounding the plant's roots as an irrigation emitter equipped with a "pocket"/bag for storing fertilizer. The objective was to evaluate the functional design of pocket fertigation in the specific micro-climate inside the screenhouse with a combination of emitter designs and irrigation rates. The technology was implemented on an experimental field at a lab-scale melon (*Cucumis melo* L.) cultivation from 23 August to 25 October 2021 in one planting season. The technology was tested at six treatments of a combination of three emitter designs and two irrigation rates. The emitter design consisted of an emitter with textile coating (PT), without coating (PW), and without emitter as a control (PC). Irrigation rates were supplied at one times the evaporation rate (E) and 1.2 times the evaporation rate (1.2E). The pocket fertigation was well implemented in a combination of emitter designs and irrigation rates (PT-E, PW-E, PT-1.2E, and PW-1.2E). The proposed technology increased the averages of fruit weight and water productivity by 6.20 and 7.88%, respectively, compared to the control (PC-E and PC-1.2E). Meanwhile, the optimum emitter design of pocket fertigation was without coating (PW). It increased by 13.36% of fruit weight and 14.71% of water productivity. Thus, pocket fertigation has good prospects in the future. For further planning, the proposed technology should be implemented at the field scale.

**Keywords:** pocket fertigation; water productivity; innovative technology; subsurface irrigation

## 1. Introduction

Irrigation and fertilization are the main components in determining agricultural production successfully. Climate change causes uncertainty in environmental conditions; thus, optimizing irrigation and fertilization should be adjusted. Suitable adaptation strategies for climate change on irrigation and fertilization could minimize the negative impacts [1]. Water resource availability tends to decrease and become more scarce with the impact of climate change [2]. However, irrigation is often oversupplied, thus resulting in more water loss and reducing water productivity [3]. In addition, excessive use of fertilizers leads to soil damage due to a large amount of soluble nitrate; thus, more nitrogen is wasted

before being absorbed by plants [4]. Therefore, it is necessary to develop water-saving and efficient technology in fertilizers. An example of water-saving irrigation technology is subsurface irrigation by the innovative emitter [5]. The technology is very effective in water use because water is supplied directly to the plant roots, reducing evaporation. Several subsurface irrigation technologies have been developed, such as ring-shaped emitter irrigation [6,7] and sheet-pipe technology [8], as well as evapotranspiration irrigation [9]. Unfortunately, the technology still does not consider the use of fertilizers yet.

Both chemical and organic fertilizers should be applied at the right time and in the right amount to avoid the loss and negative impact on the environment. The excessive use of chemical fertilizers and residue in the soil changes the soil's physical and chemical properties, so the soil is easily eroded due to decreased organic content [10]. Furthermore, fertilizers dissolve in water due to rain, and irrigation can cause eutrophication of organic matter accumulation, thus reducing water quality [11]. In addition, long-term use of chemical fertilizers causes a decrease in soil pH [12]. On the other hand, organic fertilizer is more environmentally friendly. However, it is suspected to reduce production, convincing the farmer to consider using it less [13]. In addition, a large amount of organic fertilizer content in the rainwater can make a loss in the nitrate content before being absorbed properly by the crops [4].

This study examines pocket fertigation technology as an innovative idea for water and fertilizer applications. It is developed from a previous emitter irrigation called ring-shaped subsurface irrigation [6,7,14]. This technology uses a ring/disc porous material installed surrounding the roots as an emitter and equipped with a "pocket" for fertilizer storage on the upper side. It is simple, inexpensive, effective, efficient, easy, and fast to construct and manageable by the farmers. All materials used should be available in the local markets and reachable in cost. It is in line with the "farmer-led irrigation development" program [15]. In this sense, the farmers should be capable of planning, constructing, operating, maintaining, repairing, and even developing the irrigation system. This research aims to apply such a type of irrigation technology constructible using locally available materials and easily manageable by the farmers, whether individually or collectively.

By the current technology, water is irrigated through the pocket and then flows directly to the root zone via the emitter. It is expected that water and fertilizer are absorbed by the roots simultaneously. Therefore, it is important to test the performance of the developed technology, particularly for a high economic horticultural product such as melon (*Cucumis melo* L.). Melon is a fruit that has high commercial value in Indonesia with a wide and diverse market range, from traditional markets to modern markets, restaurants, and hotels. Therefore, it can be cultivated because of its competitiveness compared to other commodities. In addition, the fruit by-product can be incubated as a functional food ingredient [16].

The current study was proposed as a preliminary study on the functional design of the pocket fertigation technology. The objective of the study was to evaluate the functional design of the pocket fertigation for melon (*Cucumis melo* L.) production particularly in the emitter design and irrigation aspect. The scope of evaluation aspects consisted of the soil moisture fluctuation, fruit weight, and water productivity under different emitters design and irrigation rates. As an indicator, soil moisture is related to water and nutrient uptake, while crop yield is related to the income obtained by the farmers [17]. In addition, water productivity is related to water use efficiency because it reflects the yield or biomass produced per water used [18].

## 2. Materials and Methods

### 2.1. Time, Location, and Soil Properties

The current preliminary study was conducted at lab-scale inside a screenhouse located at Kinjiro Farm with coordinates 6.59° S, 106.77° E, Bogor, West Java, Indonesia. *Glamor*, a variety of melon seeds, was sown on 6 August 2021, planted on 23 August 2021, and harvested on 25 October 2021. The physical characteristics of soils are presented in Table 1.

**Table 1.** The physical characteristics of planting media soil.

| No | Parameter | Value | Unit |
|---|---|---|---|
| 1 | Dry bulk density | 0.77 | g/cm$^3$ |
| 2 | Particle density | 1.92 | g/cm$^3$ |
| 3 | C-organic | 5.73 | % |
| 4 | Organic content | 9.89 | % |
| 5 | Permeability | 5.18 | cm/hour |
| 6 | Soil texture | | |
| | Sand | 17 | % |
| | Silt | 59 | % |
| | Clay | 24 | % |
| | Soil Texture | Silt Loam | |
| 7 | Soil water content at the following soil suction: | | |
| | pF 1 | 0.476 | cm$^3$/cm$^3$ |
| | pF 2 | 0.369 | cm$^3$/cm$^3$ |
| | pF 2.54 | 0.294 | cm$^3$/cm$^3$ |
| | pF 4.2 | 0.182 | cm$^3$/cm$^3$ |

Based on the physical characteristics of the soil, especially the data on soil water content at various pF (soil-water matrix potential) values, a water retention curve was made to determine the saturated and residual soil water contents by the following equation [19]:

$$\theta = \theta_r + \frac{(\theta_s - \theta_r)}{[1 + (\alpha h)^n]^m} \tag{1}$$

where $\theta$ is the soil moisture (m$^3$/m$^3$) in volumetric water content, $\theta_s$ is the saturated soil water content (m$^3$/m$^3$), $\theta_r$ is the residual soil water content (m$^3$/m$^3$), h is the pressure head (cm H$_2$O), and $\alpha$, n, and m are constants. The values of $\theta_s$, $\theta_r$, $\alpha$, n, and m were optimized with a solver in Microsoft Excel (Figure 1). From the optimization results, the values of $\theta_s$ and $\theta_r$ were 0.485 m$^3$/m$^3$ and 0.100 m$^3$/m$^3$, respectively.

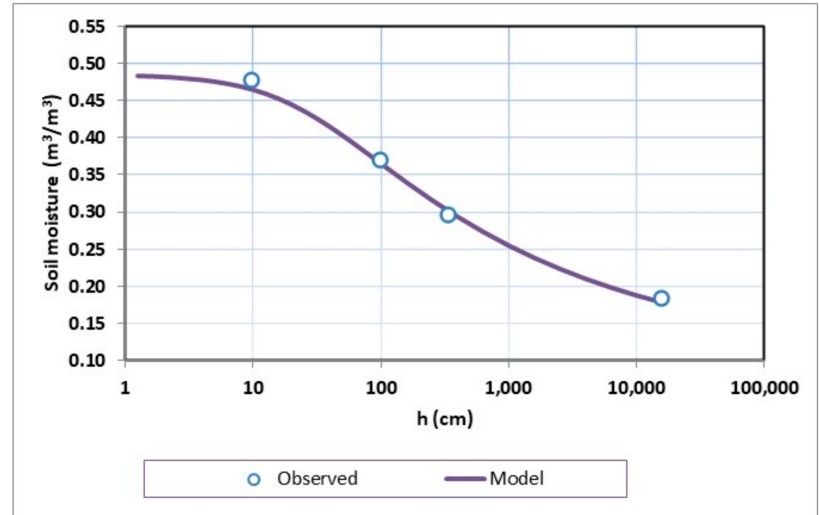

**Figure 1.** Water retention curve for the type of soil at the study site.

## 2.2. Experimental Design of the Pocket Fertigation

The experimental design consisted of a combination of emitter types of the pocket fertigation and irrigation rates with six treatments and two replications in total. The pocket fertigation was applied in a pot experiment with a 50 cm diameter in the top and 30 cm diameter in the bottom (Figure 2a). Meanwhile, the design of pocket fertigation is presented

in Figure 3. Here, two designs were developed with the same dimensions. As previously mentioned, pocket fertigation has two parts: an emitter and a pocket to store the fertilizer. The emitter material was made from a perforated hose, 14 holes in total, with the interval of the hole being 5 cm. The first design of the emitter was coated with a textile material (PT) and without coating material (PW). The emitter was oval with a longer diameter of 30 cm and a shorter one of 25 cm. The pocket's diameter was 9 cm with a 25 cm height that was created from used plastic bottles with a size of 1500 mL. In this experiment, the emitter was placed 5 cm below the soil surface. For the control, surface irrigation was applied in which the fertilizer was sprinkled on the soil surface (PC).

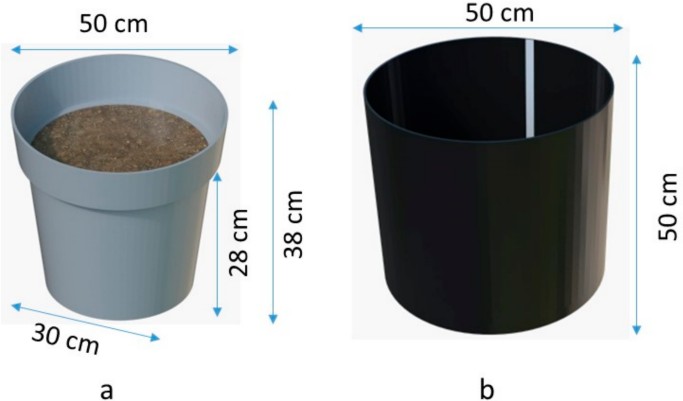

**Figure 2.** (**a**) The dimensions of pot; (**b**) the dimensions of pan evaporation.

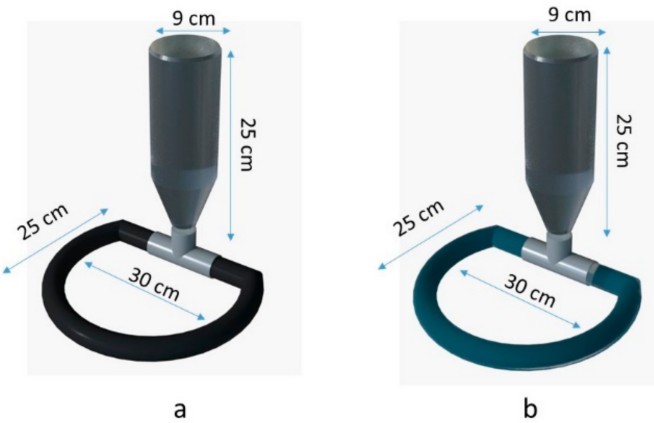

**Figure 3.** The pocket fertigation design: (**a**) emitter with textile coating (PT), (**b**) emitter without coating (PW).

For the irrigation rate, it is commonly supplied based on crop evapotranspiration (ETc); however, it is difficult to apply by the farmer due to the complicated method. In this research, we used a simple method by pan evaporation to determine the open water evaporation rate on a daily basis. The irrigation water was supplied based on the evaporation rate, i.e., one times the evaporation (E) and 1.2 times the evaporation (1.2E) in all designs of emitters, so there were six treatments in total, i.e., PT-E, PW-E, PC-E, PT-1.2E, PW-1.2E, and PC-1.2E (Figure 4). For the pan evaporation, we used a pan filled with water, 50 cm in diameter and height (Figure 2b). The daily evaporated water was recorded every morning (around 7.00 a.m.). For the leaching process, all treatments were supplied with more water ranging from 2 to 4 L/plant six times at 26, 33, 38, 41, 46, and 51 days after transplanting (DAT). In addition, this watering was also performed to avoid extreme drought in the growing media.

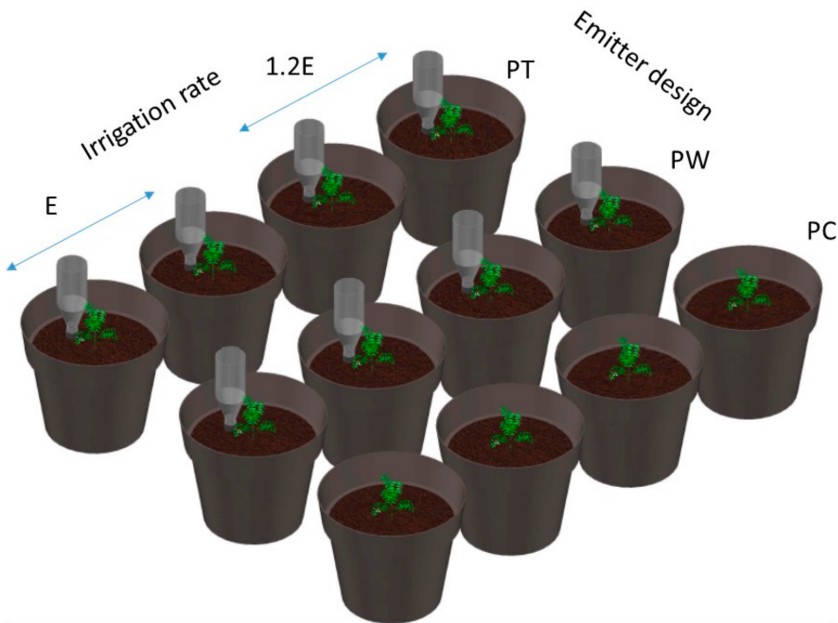

**Figure 4.** Testing of the pocket fertigation with various emitter designs and irrigation rates.

As we focused on the application of pocket fertigation under different irrigation rates, during the experiment, all treatments were given the same amount and materials content of fertilizer. They were "ABmix" and NPK "Mutiara" fertilizers. The "ABmix" fertilizer contains macro and micro-nutrients. During the planting season, the "ABmix" fertilizer was dissolved with an EC (Electrical Conductivity) value of 4500–5000 μS/cm and the NPK "Mutiara" fertilizer of 20 g/plant at 20 DAT was stored in the pocket.

### 2.3. Micro-Climate and Soil Moisture Monitoring

The micro-climate inside the screenhouse was measured by an automatic weather station (AWS) connected to the server. It was part of an IoT-based measurement previously developed [20]. There were several weather sensors, i.e., air temperature, relative humidity, wind speed, and solar radiation. Each parameter was measured at 15 min intervals. The micro-climate conditions in the screenhouse fluctuated throughout the cultivation period. However, the daily average, minimum, and maximum air temperatures had a constant trend (Figure 5a). The daily minimum, average, and maximum air temperature values ranged between 22 °C, 28 °C, and 35 °C, respectively. The same thing also occurred with the relative humidity (RH). Although it fluctuated more, the trend was also relatively constant with the average value of RH being approximately was 82% (Figure 5a). Something quite extreme happened on 14 September 2021 (22 DAT). The daily maximum and average air temperatures decreased significantly. On the other hand, RH increased significantly. Here, the daily maximum temperature only reached 27.7 °C with an average of 25.1 °C. Meanwhile, the RH increased and reached a maximum value of 90.1%. In atmospheric pressure, air temperature and RH are inversely proportional, as presented in Figure 5b. The type of greenhouse strongly influences variations in air temperature and RH in the greenhouse used [21]. The air temperature inside the greenhouse should be controlled properly because an increase in air temperature before harvest can reduce fruit sweetness [22]. Many air temperature control systems, including RH control systems, have been developed for optimal plant growth, such as fuzzy control systems [23,24].

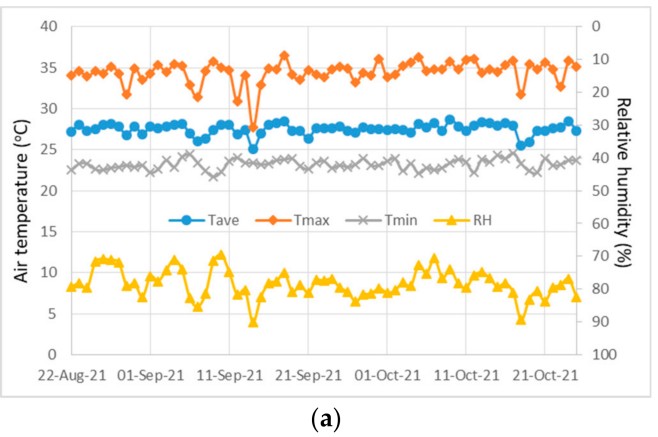
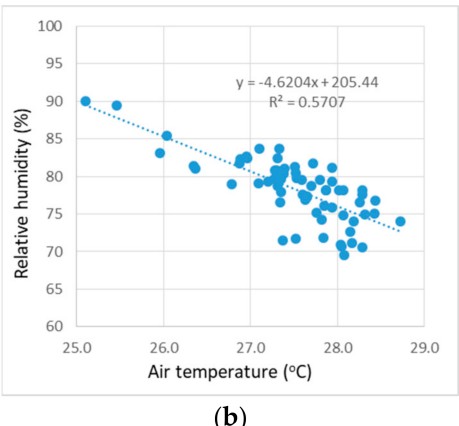

(a)                                                                        (b)

**Figure 5.** (**a**) Daily maximum, average, and minimum air temperatures, and relative humidity; (**b**) linear correlation between daily average air temperature and relative humidity.

The weather data (air temperature, relative humidity, wind speed, and solar radiation) were then used to determine the reference evapotranspiration based on the following Penman–Monteith equation [25]:

$$ET_o = \frac{0.408\Delta(R_n - G) + \gamma \frac{900}{T_{ave}+273} u(e_s - e_a)}{\Delta + \gamma(1 + 0.34u)} \tag{2}$$

where ETo is the reference evapotranspiration (mm), Rn is the net radiation ($MJ/m^2/d$), G is the soil heat flux density ($MJ/m^2/d$), $T_{ave}$ is the daily average air temperature (°C), u is the wind speed (m/d), $e_s$ is the saturated vapor pressure (kPa), $e_a$ is the actual vapor pressure (kPa), $\gamma$ is the psychrometric constant (kPa/°C), and $\Delta$ is the slope of the vapor pressure curve (kPa/°C). Rn, G, $e_s$, $e_a$, and $\gamma$ were determined based on observed solar radiation and relative humidity parameters. In addition, to perform the equation, elevation, latitude, and Julian day data were required. The data were compared to evaporation rate that was measured daily as previously explained.

For effectiveness of emitter design, the soil moisture was monitored at a depth of 5 cm below the soil surface and in the middle of the emitter. The 5-TE soil moisture sensor from the Meter Group was used for this purpose. The sensor was placed at a 5 cm soil depth because the emitter of pocket fertigation was kept at this location. The sensor was connected to a ZL datalogger (Meter Group) with a measurement interval of 15 min. From the fluctuations in soil moisture, the actual evapotranspiration between the treatment was estimated and compared.

*2.4. Crop Performances and Water Productivity Analysis*

The indicators of crop performance were plant growth, fruit weight, and soluble solid content. The soluble solid content represented the sweetness level of fruit. For plant growth parameters, the number of leaves and plant height were measured at the ages of 10, 20, and 30 DAT during the vegetative phase. Meanwhile, in the generative phase (fruit formation), fruit weight and total soluble solid content representing sweetness levels were observed on the harvesting day. The total soluble solid was measured by the Atago Pocket Digital Refractometer in % Brix.

Water productivity was determined based on the product produced per amount of water used based on the definition [26]. As the experiment was conducted inside a screen house and there was no rain, the equation for water productivity is represented as follows:

$$WP_I = \frac{Y}{I}C \tag{3}$$

where Y is the fruit weight (g), I is the total irrigation (mL), C is the conversion factor (in this case, 1000), and $WP_I$ is the water productivity based on total irrigation water (kg weight/m$^3$ water).

### 2.5. The Limitation of the Study

The current study only presented the functional design of pocket fertigation. The evaluation scopes were on soil moisture fluctuation, evapotranspiration, and crop and water productivities. As the numbers of pots and screenhouse areas were limited, statistical analysis was limited on the average value and standard deviation. Thus, the values will be compared among the treatments. The proposed technology will be implemented at field scale and it is planned for the next phase of the study.

## 3. Results

### 3.1. Evaporation and Evapotranspiration during the Season

Figure 6 shows fluctuations in solar radiation, evaporation, and reference evapotranspiration (ETo) during the growing season. Inside the screenhouse, the solar radiation was relatively low, ranging from 2.1 to 9.8 MJ/m$^2$/d. The low solar radiation affected the low reference evapotranspiration and pan evaporation (Figure 6). The reference evapotranspiration value ranged from 0.4 to 2.2 mm, while the pan evaporation was from 1 to 4 mm. The pan evaporation value was higher than the reference evapotranspiration because more water evaporated from the water surface than in the soil media when the soil was unsaturated, as found in all treatments. This condition is in line with previous experiments that stated that evaporation increases with the presence of flooded water (unsaturated condition) in the soil and vice versa [27].

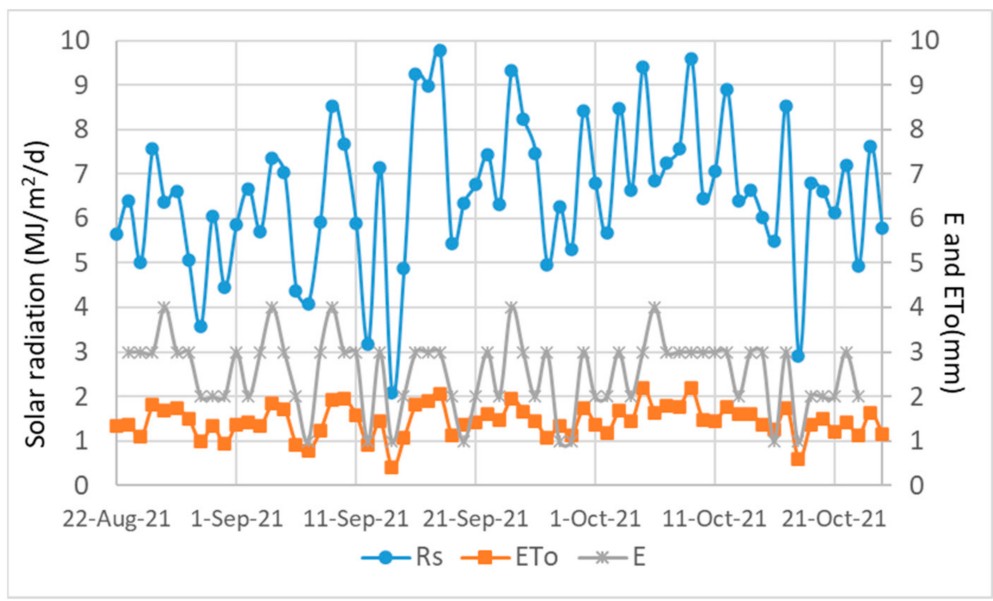

**Figure 6.** Daily total solar radiation, evaporation (E), and reference evapotranspiration (ETo).

ETo was strongly correlated with solar radiation, represented by high R$^2$ (>0.85), as shown in Figure 7a. Therefore, solar radiation is the strongest parameter affected on the ETo [20]. The minimum ETo was 0.4 mm when the solar radiation was also a minimum (2.1 MJ/m$^2$/d). A similar condition also existed for its maximum value, which reached 2.2 mm when the solar radiation was at its maximum level (9.8 MJ/m$^2$/d). It was indicated that solar radiation had the greatest influence on the evapotranspiration process particularly through the soil surface and plants [28]. The solar radiation also had a positive correlation to evaporation, although it had a lower R$^2$ compared to the ETo correlation (Figure 7b). Evaporation also correlated (R$^2$ > 0.48) to ETo, as shown in Figure 7c. It was indicated that evaporation from the water surface and evapotranspiration (evaporation and transpiration)

occurred simultaneously. Commonly, evaporation from the water surface ($E_{pan}$) was higher than that of evaporation from the soil surface, which was measured by a lysimeter ($E_{lys}$) [29]. Evaporation can be converted to evapotranspiration via the pan coefficient (Kp) [30]. In this study, based on empirical data, Kp was 0.56, indicating that evaporation was approximately 56% higher than the ETo.

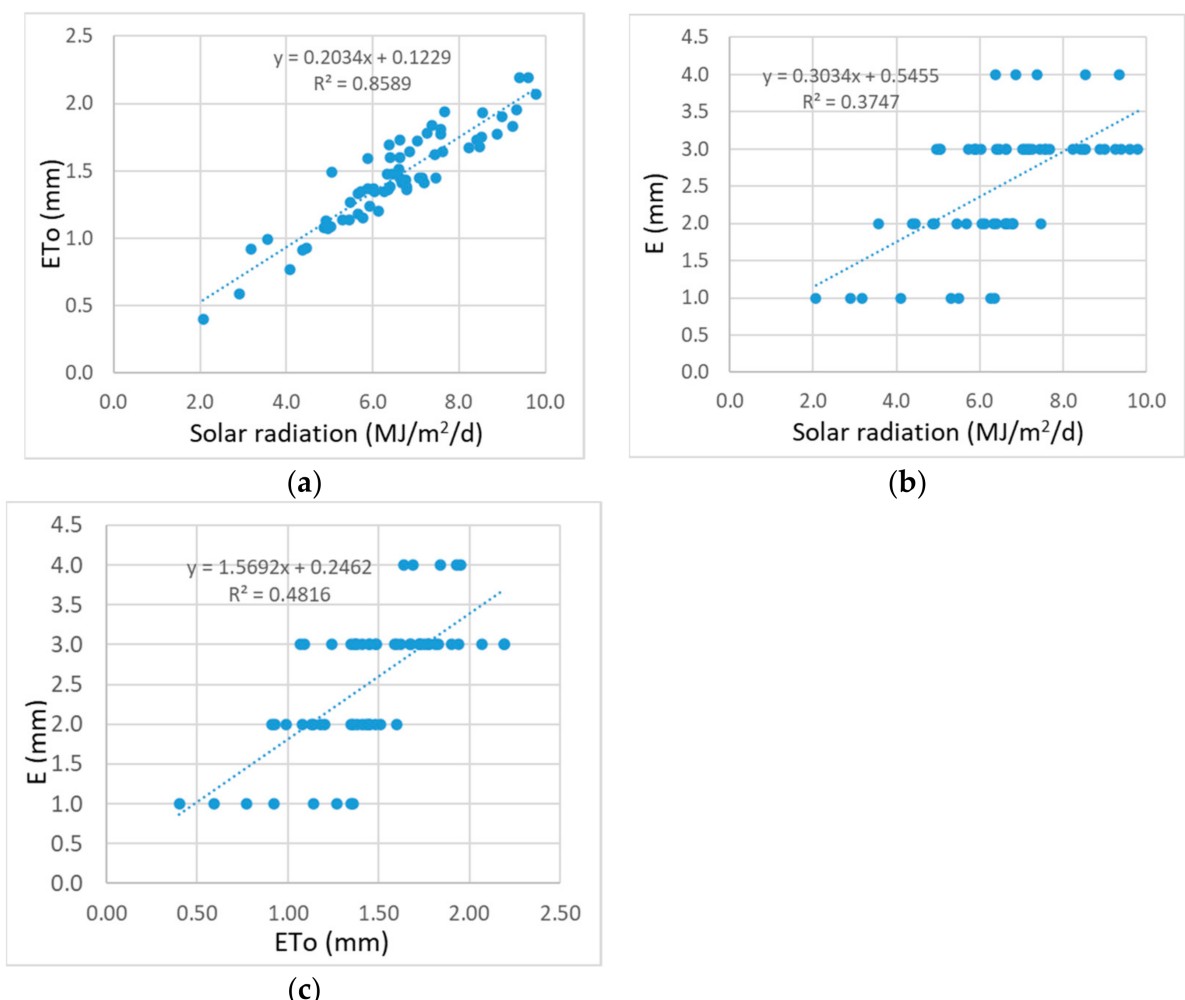

**Figure 7.** Relationship between (**a**) reference evapotranspiration (ETo) and solar radiation, (**b**) evaporation (E) and solar radiation, and (**c**) evaporation (E) and reference evapotranspiration (ETo).

### 3.2. Soil Moisture Conditions in Various Irrigation Rates

For in-plant cultivation systems inside the screenhouse or greenhouse, soil moisture is the key to success in horticultural crop production. Thus, it is important to control soil moisture accurately [31]. The soil moisture in PT-E and PT-1.2E fluctuated depending on the irrigation supplied because the plant water requirement for the plants was only supplied from irrigation (Figure 8). The PT-1.2E with a higher irrigation rate had higher soil moisture levels than those in the PT-E. At the PT-1.2E, soil moisture ranged from 0.198 to 0.496 m³/m³, while at PT-E, it ranged from 0.116 to 0.437 m³/m³. The highest soil moisture level occurred at 41 DAT (3 October 2021) when 4000 mL of irrigation was supplied to PT-E and PT-1.2E treatments. At this time, the soil moisture value was reached at its saturation level in the PT-1.2E. However, the maximum soil moisture in the PT-E treatment was still lower than that of the soil saturation level. At both irrigation rates (E and 1.2E), the soil moisture tended to be at the field capacity level at the beginning of the vegetative phase. Then, water irrigation in large quantities was supplied when the soil moisture level was

too low, particularly in the mid-season phase. In the generative phase, the soil moisture condition was maintained in the range of field capacity in both irrigation rates.

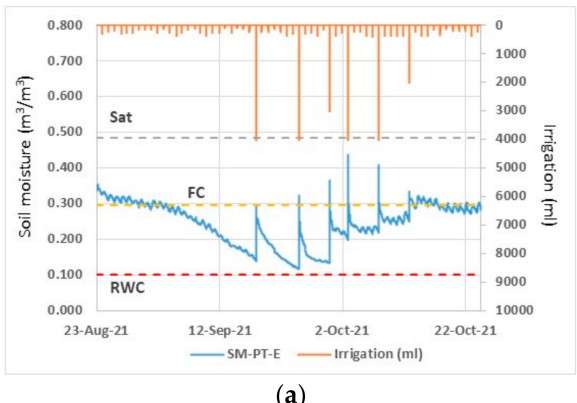

(**a**)

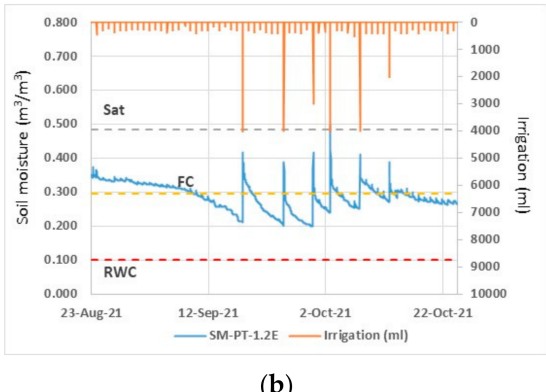

(**b**)

**Figure 8.** The fluctuation in soil moisture and irrigation: (**a**) PT-E treatment, (**b**) PT-1.2E treatment. Note: Sat: saturated water content, FC: field capacity water content, RWC: residual water content.

A similar thing occurred with the PW treatments (Figure 9). The soil moisture level increased rapidly when a large amount of irrigation was supplied. In the PW-E, soil moisture was slightly higher than the field capacity level in the beginning phase until 6 DAT (29 August 2021). Then, the soil moisture decreased below field capacity level until harvest. Here, the soil moisture conditions ranged from 0.147 to 0.339 $m^3/m^3$. Meanwhile, as more water was supplied, soil moisture in the PW-1.2E was consequently higher than that in the PW-E. At the beginning phase, the soil moisture was at field capacity level until 23 DAT (15 September 2021), and it reached the saturation level when a large amount of water was supplied, particularly at 26 DAT. Hereafter, soil moisture was below the field capacity level. In this treatment, soil moisture ranged from 0.150 to 0.493 $m^3/m^3$. Overall, the average soil moisture in the PW-E and PW-1.2E was 0.222 and 0.269 $m^3/m^3$, respectively.

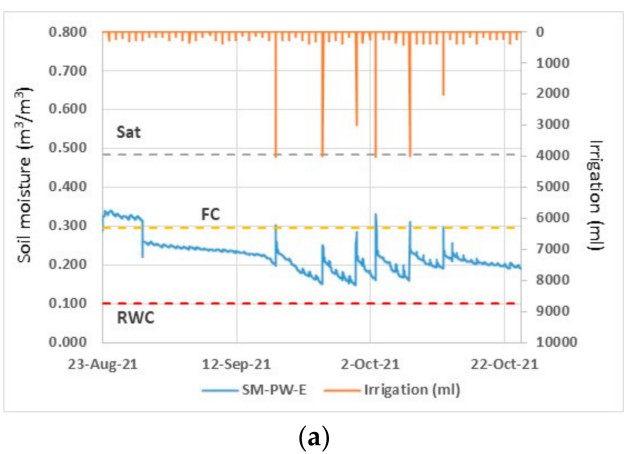

(**a**)

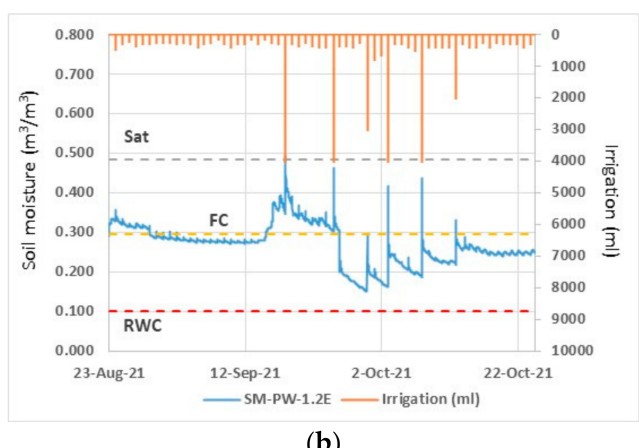

(**b**)

**Figure 9.** The fluctuation in soil moisture and irrigation: (**a**) PW-E treatment, (**b**) PW-1.2E treatment. Note: Sat: saturated water content, FC: field capacity water content, RWC: residual water content.

The fluctuations in soil moisture of the PC treatments are presented in Figure 10. The soil moisture level in the PC-E ranged from 0.112 to 0.426 $m^3/m^3$, while at the PC-1.2E, it ranged from 0.135 to 0.454 $m^3/m^3$. For the PC-E, the soil moisture level was below the field capacity level for most of the growing period, except on the specific days (at 26, 33, and 41 DAT) when large amounts of water were applied. Meanwhile, in the PC-1.2E, soil moisture ranged from the field capacity level in the beginning phase to 26 DAT, and then dropped to below field capacity.

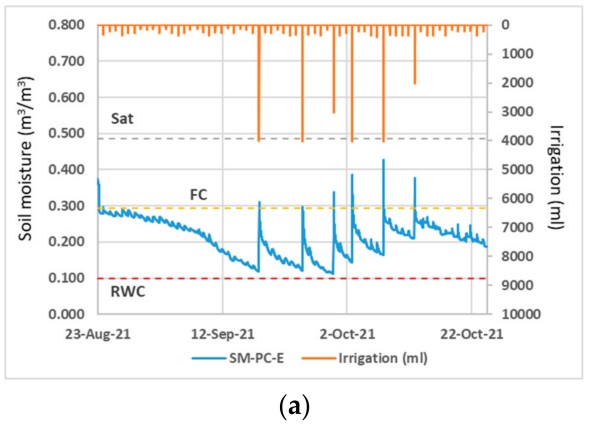
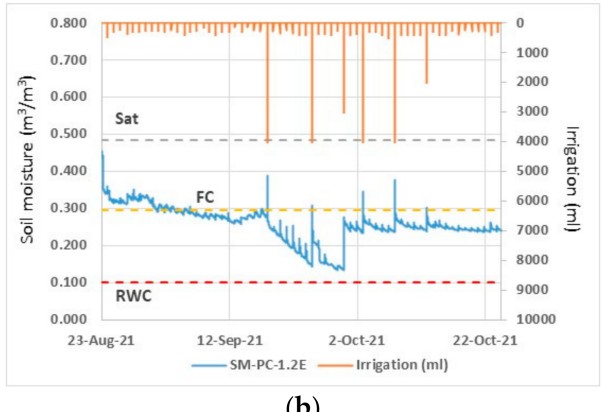

(**a**)                                                                                              (**b**)

**Figure 10.** The fluctuation in soil moisture and irrigation: (**a**) PC-E treatment, (**b**) PC-1.2E treatment. Note: Sat: saturated water content, FC: field capacity water content, RWC: residual water content.

Table 2 shows the average value of soil moisture levels for each treatment every 10 DAT. Among the two emitter designs (PT and PW) of pocket fertigation, soil moisture tended to be stable with an average level close to the field capacity. PW was more able to maintain soil moisture above the value of 0.200 m$^3$/m$^3$ compared to PT. This means the emitter without the coating distributed irrigation water more uniformly and it also reduced actual evapotranspiration by up 13.6%. This indicated that the PW was probably more efficient in water use compared to PT.

**Table 2.** The maximum, average, minimum soil moisture, and actual evapotranspiration among the treatments.

| Parameters | Treatments | | | | | | Summary | |
|---|---|---|---|---|---|---|---|---|
| | PT-E | PT-1.2E | PW-E | PW-1.2E | PC-E | PC-1.2E | Pocket Fertigation * | Control ** |
| Soil moisture (m$^3$/m$^3$) at: | | | | | | | | |
| 0–10 (DAT) | 0.312 | 0.334 | 0.295 | 0.306 | 0.276 | 0.326 | 0.312 | 0.301 |
| 11–20 (DAT) | 0.257 | 0.304 | 0.240 | 0.277 | 0.223 | 0.283 | 0.269 | 0.253 |
| 21–30 (DAT) | 0.181 | 0.263 | 0.214 | 0.330 | 0.158 | 0.250 | 0.247 | 0.204 |
| 31–40 (DAT) | 0.170 | 0.238 | 0.178 | 0.228 | 0.153 | 0.187 | 0.203 | 0.170 |
| 41–50 (DAT) | 0.245 | 0.293 | 0.202 | 0.225 | 0.216 | 0.252 | 0.241 | 0.234 |
| 51–62 (DAT) | 0.293 | 0.282 | 0.207 | 0.252 | 0.225 | 0.245 | 0.258 | 0.235 |
| Maximum | 0.312 | 0.334 | 0.295 | 0.330 | 0.276 | 0.326 | 0.334 | 0.326 |
| Minimum | 0.170 | 0.238 | 0.178 | 0.225 | 0.153 | 0.187 | 0.170 | 0.153 |
| Average | 0.243 | 0.285 | 0.223 | 0.270 | 0.209 | 0.257 | 0.255 | 0.233 |
| ETa (mm) | 118.8 | 123.9 | 98.7 | 114.9 | 143.9 | 107.9 | 114.1 | 125.9 |

* average value of PT-E, PT-1.2E, PW-E, PW-1.2E. ** average value of PC-E and PC 1.2E.

Table 2 also shows that the pocket fertigation was better than the control treatment in retaining soil moisture at a depth of 5 cm. The indicator had a higher soil moisture at the pocket fertigation than that of the control treatment. In addition, pocket fertigation was able to reduce the actual evapotranspiration by 10.32% of the control. The pocket fertigation functioned well, indicated by the higher efficiency of water used. It was seemingly subsurface irrigation that was more effective in distributing water along the root zone than that of surface irrigation. Previous research utilizing a similar emitter type showed that subsurface irrigation can maintain soil moisture in the root zone without causing stress to the plants [7].

### 3.3. Plant Growth and Their Productivities

The vegetative growth in each treatment is depicted in Figure 11. The highest average number of leaves at 20 DAT was produced by the PT, followed by the PC and PW treatments. However, the PW grew the highest plant height at 20 DAT, followed by the PT and PC treatments. After 30 DAT, pruning of the plants was carried out by maintaining the height of each plant at 200 cm. Overall, the vegetative growth among the treatments was comparable, particularly after 30 DAT.

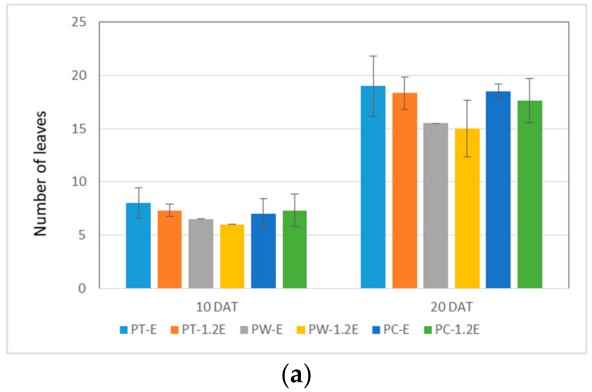

(**a**)

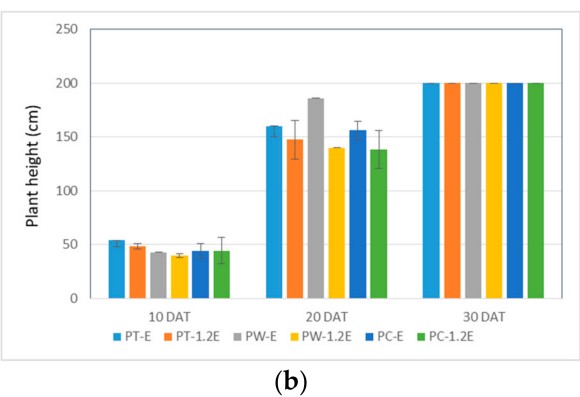

(**b**)

**Figure 11.** Plant growth performances among the treatments: (**a**) number of leaves, (**b**) plant height.

According to Table 3, the PW produced a 13.36% bigger average fruit weight than that of the PT. However, the PW produced a 13.07% lower total soluble solid than that of the PT. From the perspective of water used, the PW was more efficient as represented by the higher water productivity by up 14.71%. Therefore, it is recommended to use the pocket fertigation without coating materials. The lower effectivity of the PT is probably due to the clogging problems that occurred by the sedimentation of fertilizers. Thus, this clogging inhibited the distribution of water and fertilizer in the root zone. Clogging is generally a problem that must be overcome when utilizing irrigation systems with low flow rates [32], such as subsurface irrigation.

**Table 3.** Crop and water productivities among the treatments.

| Treatments | Yield (Fruit Weight) (g) | Irrigation (mL) | Total Soluble Solid (%brix) | WP (kg/m$^3$) |
|---|---|---|---|---|
| PT-E | 733 ± 50.9 | 34,825 | 10.5 ± 0.0 | 21.0 |
| PT-1.2E | 925.5 ± 116.7 | 38,925 | 9.4 ± 2.4 | 23.8 |
| PW-E | 898 ± 0 | 34,825 | 9.3 ± 0 | 25.8 |
| PW-1.2E | 982 ± 5.7 | 38,975 | 8.3 ± 0.5 | 25.2 |
| PC-E | 551 ± 0 | 34,875 | 10.8 ± 0 | 15.8 |
| PC-1.2E | 1115 ± 0 | 38,925 | 7.7 ± 0 | 28.6 |
| Pocket Fertigation | 885 ± 92.6 | 36,888 | 9.4 ± 0.8 | 24.0 |
| Control | 833 ± 282.0 | 36,900 | 9.3 ± 1.5 | 22.2 |
| Irrigation rate at E | 727.3 ± 141.7 | 34,842 | 10.2 ± 0.6 | 20.9 |
| Irrigation rate at 1.2E | 1007.5 ± 79.4 | 38,942 | 8.5 ± 0.7 | 25.9 |

Note: The presented data are the mean ± SD.

Table 3 shows that better performances were found in the pocket fertigation for fruit weight, total soluble solid, and water productivity compared to the control. It increased the average fruit weight by 6.20% and water productivity by 7.88%. Meanwhile, a higher water irrigation rate at 1.2E produced a bigger fruit weight than that at the E irrigation rate. Fruit weight increased significantly by 38.53% (Table 3). The increasing fruit weight of 1.2E may be contributed by increasing the actual evapotranspiration due to more irrigation water, particularly in the pocket fertigation (Table 2). This reason was supported by a previous study [33]. However, the increase in the fruit weight decreased the sweetness level (total

soluble solid), as shown in Figure 12. The heavier melon, the higher water content, and the low dissolved solids may reduce the sweetness level. The results are similar to the previous observation [34,35].

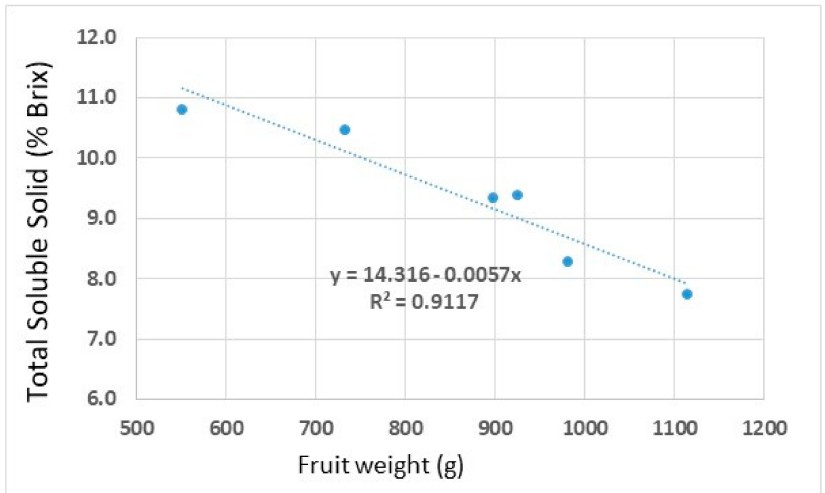

**Figure 12.** Relationship between total soluble solid and weight of fruit.

## 4. Discussion

In the context of climate change, water resources for the agriculture sector may become scarce in the future. Therefore, it is important to develop innovative and applicable technologies in utilizing irrigation water more effectively and efficiently, such as the pocket fertigation. Pocket fertigation is easy to produce by the farmers in Indonesia. The basic materials are a hose as the emitter and used bottles to store the fertilizer. In this preliminary study with a limited area, the pocket fertigation was shown to retain soil moisture better than surface irrigation as a control. Maintaining soil moisture implies that more water is stored in the soil, and it can be utilized by plants more optimally. Consequently, the fruit weight was heavier and had higher water productivity (Table 3).

The irrigation water delivery method of the pocket fertigation is similar to drip irrigation in which the emitter is placed below the soil surface near the root zone. Subsurface irrigation, both the pocket fertigation and drip irrigation, proved to be more effective and efficient in the utilization of irrigation water by reducing water loss due to evapotranspiration, as shown in Table 3 and reported in previous studies. As reported by Wang et al. [36], a long-time field experiment of drip irrigation in 2014–2018 showed that irrigation reduced 0.1–23% of evaporation and 7% of evapotranspiration per year. Consequently, the water use efficiency of drip irrigation can be significantly improved under various crop evapotranspiration scenarios [37]. In addition, subsurface irrigation with drip irrigation, combined with fertigation, increased production up to 41% as reported by Rolbiecki et al. [38]. Subsurface irrigation is not only known as effective and efficient in water used, but also more environmentally friendly. The indicates a reduction in greenhouse emissions from the soil under subsurface irrigation, especially $N_2O$ and $CO_2$ [39,40].

The current developed technology has good prospects in the near future and should be continuously developed. Pocket fertigation is a kind of subsurface irrigation. It has a better performance indicated by the higher effectiveness of water use, and consequently, it can increase water productivity [31]. The performance tests on a field scale are needed not only for melon (*Cucumis melo* L.) but also for other crops. Crop type selection depends on the local climate condition and farmer's preference. Several locations in Indonesia are characterized by dry areas with low rainfall intensity such as East Nusa Tenggara (NTT), a province located in eastern Indonesia [41]. The location lacks water resources, so it is very appropriate to be chosen as the location for field-scale trials.

## 5. Conclusions

An innovative technology, pocket fertigation, was well implemented in the lab-scale experiment. The pocket fertigation with subsurface irrigation was better than surface irrigation in retaining soil moisture at a 5 cm soil depth. The soil moisture could be maintained at nearly field capacity level. The pocket fertigation was able to reduce the actual evapotranspiration by 10.32%. It also showed better performances in fruit weight production and water productivity. It increased the average fruit weight by 6.20% and water productivity by 7.88%, respectively. Thus, pocket fertigation has good prospects in the future. For further planning, the proposed technology will be implemented at the field scale, particularly in dry areas with minimum water resources.

**Author Contributions:** Conceptualization, C.A. and B.I.S.; methodology, C.A. and B.I.S.; data collection, C.A., A.M. and S.F.D.S.; writing—original draft preparation, C.A.; writing—review and editing, C.A., B.I.S., Y.W., B.D.A.N., M.M. and A.A. All authors have read and agreed to the published version of the manuscript.

**Funding:** This research was funded by the Indonesian Collaborative Research Program—WCU (World Class University) scheme by IPB University for the 2021 fiscal year with the number 1376/IT3.L1/PN/2021 dated 23 February 2021 by the project title "Developing Innovative Pocket Fertigation Technology based on Artificial Intelligence and Adaptive to Climate".

**Institutional Review Board Statement:** Not applicable.

**Informed Consent Statement:** Not applicable.

**Data Availability Statement:** The data presented in this study are available upon request from the corresponding author. The data are not publicly available, due to shared ownership between all parties that contributed to the research.

**Acknowledgments:** We would like to thank and appreciate to reviewers for all valuable comments, critics and suggestions, which helped us to improve the quality of the article. Also, we thank to Ahmad Kohar and Ibrahim for helping us in the field.

**Conflicts of Interest:** The authors declare no conflict of interest.

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
