# Peer review of "Functional Design of Pocket Fertigation under Specific Microclimate and Irrigation Rates: A Preliminary Study"

_agronomy, doi:10.3390/agronomy12061362_

Round 1

Reviewer 1 Report

General comment:

The manuscript presented a study on the performances of an innovative drip emitter called pocket fertigation and its effect on crop and water productivities of melon cultivation under different irrigation regimes. The manuscript tried to address the challenges faced by agricultural production from the threat of over watering and traditional fertilization, which can be translated into climate change and environmental pollution. The work can be justified based on the current socioeconomical and environmental challenges. However, the study was poorly designed and could not achieve the stated aims. There was so much emphasis on the microclimate though it was not part of the article objectives. Based upon the above and the following specific comments, I recommend resubmitting.

Other Comments:

  1. The experiment was not well designed. The name of the design was not clearly stated and obviously there was no replications of treatments.
  2. From Figure 3 – it Seems the pots were not randomized. Also, what is the meaning of “1.5E” used here? The size of the pot is not stated.
  3. Water used by crop should be replaced based on evapotranspiration which could have been estimated with the pan evaporation. However, the method used in this article did not replace crop ET but rather applied certain amount based on evaporation.
  4. Line 156- The moisture content monitoring was limited to 5 cm only, which will not represent the soil moisture variation of the entire rooting zone or majority of it.
  5. Line – 162 The details sampling procedure for yield determination was not provided.
  6. Line 169- Water productivity is measured in either kg/m3 or kg/cm3
  7. Line 172 – Since the statistical analysis was not conducted to separate the means, adequate conclusion cannot be made.

Reviewer 2 Report

The idea of making a mixture of nutrients and put it with water to provide the plant with water and nutrients at its request seem interesting. It could be useful for domestic use, but I find it hard to see its applicability on an industrial or even small-medium farm-scale.

Apart from this, the work in my opinion does not reach the scientific level required to be published in a journal like Agronomy. These are my main reasons for not recommending its publication:

- There is no experimental design where a hypothesis is being tested, for example that plants with pocket fertigation perform better that plants without pocket fertigation. The number of replications, blocks, ... is unknown. Therefore, the results and conclusion cannot be supported by data. This is enough to invalidates this work for publication in a scientific journal.

-The results give a very broad and detailed explanation of the climate inside the greenhouse, perhaps much of this information is unnecessary. Moreover, this information should be part of the growing conditions in the Material and Methods section.

- The results give little explanation of how the treatments affected vegetative development and fruit production. I miss data about plant biomass at the end of the experiment, data about plant water and nutrient status. To test the performance of this system (pocket fertigation) again established practices more effort is necessary in terms of plant monitoring.

Minor concerns;

-In Material and Methods, a section explaining plant material and growth conditions is missing. In this subsection, the drip irrigation system should be explained.

-English should be improved. There are some sentences difficult to understand.

This is an example of research that unfortunately could not be completed, due to problems with pests and diseases. This happens all the time to all scientists, is part of our job. When this happens we need to repeat the experiment. I recommend repeating it again, with a robust experimental design, and with planned measurements of soil moisture and vegetative growth and development and maybe some plant mineral analysis should be welcome. With all this, you can validate the suitability of this fertigation technique for potting’s plants.

Reviewer 3 Report

The article Performances of Pocket Fertigation on Crop and Water Productivities of Melon (Cucumis melo L) Cultivation under Different Irrigation Regimes contains an interesting experiment. The research conducted by the authors is correctly described and increases the current state of knowledge in the field of melon fertigation. The main drawback of the article is the lack of discussion of the results in relation to previous studies and world literature. Please supplement this. If there is a lack of research on this fertigation method, please at least relate the results in the context of irrigation. After significant development of the discussion of the obtained results, in my opinion, the article will be suitable for publication

Round 2

Reviewer 2 Report

For my part it is acceptable for publication, they bring to light a novel technique that will be useful to them and other researchers.